# Application of Modified mRNA in Somatic Reprogramming to Pluripotency and Directed Conversion of Cell Fate

**DOI:** 10.3390/ijms22158148

**Published:** 2021-07-29

**Authors:** Aline Yen Ling Wang

**Affiliations:** Center for Vascularized Composite Allotransplantation, Chang Gung Memorial Hospital, Taoyuan 333, Taiwan; aline2355@yahoo.com.tw

**Keywords:** modified mRNA, induced pluripotent stem cells, mRNA-based reprogramming, transdifferentiation, therapeutic application

## Abstract

Modified mRNA (modRNA)-based somatic reprogramming is an effective and safe approach that overcomes the genomic mutation risk caused by viral integrative methods. It has improved the disadvantages of conventional mRNA and has better stability and immunogenicity. The modRNA molecules encoding multiple pluripotent factors have been applied successfully in reprogramming somatic cells such as fibroblasts, mesenchymal stem cells, and amniotic fluid stem cells to generate pluripotent stem cells (iPSCs). Moreover, it also can be directly used in the terminal differentiation of stem cells and fibroblasts into functional therapeutic cells, which exhibit great promise in disease modeling, drug screening, cell transplantation therapy, and regenerative medicine. In this review, we summarized the reprogramming applications of modified mRNA in iPSC generation and therapeutic applications of functionally differentiated cells.

## 1. Introduction

Embryonic stem cells (ESCs) are derived from the inner cell mass (ICM) of embryos at the early stage of blastocyst development [1]. ESCs are pluripotent stem cells and are able to differentiate into any cell type of the body. Thus, pluripotent stem cells hold promise in regenerative medicine and disease modeling. However, harvesting the embryoblast results in the destruction of the blastocyst and therefore faces ethical issues. In 2006, Yamanaka et al. first discovered and generated induced pluripotent stem cells (iPSCs) from mouse fibroblasts transduced with four transcription factors such as Oct3/4, Sox2, c-Myc, and Klf4 (OSKM), under ES cell culture conditions [2]. These cells exhibit characteristics similar to ESCs such as morphology, growth, pluripotency, and marker expression [3,4]. ESCs are derived from the ICM of the blastocyst, whereas iPSCs are derived from a variety of somatic cell types using various reprogramming techniques. Unlike ESCs, iPSCs are more readily obtainable for therapy and research, and their harvest does not face ethical concerns [5]. For example, the mutated genes of iPSCs derived from diseased patients can be repaired using the homologous recombination method and gene editing using CRISPR–Cas9 technology. The gene-corrected terminally differentiated cells derived from diseased iPSCs can be transferred into diseased patients for cell therapy. iPSCs generated from healthy or diseased cells can also be used for the in vitro screening of drug candidates [6]. Therefore, autologous iPSCs-derived therapeutic cells are preferred for use in diseased patients. This shows great promise for precision and personalized medicine.

There are four major reprogramming methods for the generation of iPSCs derived from somatic cells through the overexpression of pluripotent facts. (1) Virus-based integrative reprogramming method: virus transduction of reprogramming factors such as retrovirus and lentivirus were used for overexpression of Oct3/4, Sox2, c-Myc, and Klf4 in somatic cells, which lead to somatic cell transformation into pluripotent stem cells. Although virus-based methods lead to highly efficient reprogramming cells (0.01~0.1%) [7,8,9], virus-carrying transgenes would be randomly integrated into chromosomes of somatic cells. This causes iPSCs to be at risk of chromosome gene mutation. The use of proto-oncogene c-Myc in gene reactivation could increase the risk of transgene-derived tumor formation. Although cure-excisable lentiviral systems offer a solution to genome integration, they require lengthy sub-cloning procedures and screening to ensure excision of the reprogramming factors. (2) RNA virus-based non-integrative reprogramming method: Sendai virus is used for the overexpression of pluripotent transgenes and subsequently iPSC generation. The Sendai virus method is easy to use and leads to highly efficient reprogramming cells (0.01~1%) [10,11,12], whereas residual Sendai virus is difficult to clear from somatic cells. This needs several rounds of clonal expansion and analysis. (3) Plasmid-based non-integrative reprogramming method: Episomal DNA plasmid-carried pluripotent genes are used for iPSC generation, which are transgene free [13,14]. The technique was gradually improved, and the reprogramming efficiency ranged between 0.04% and 0.3% [13]. The elimination of residual episomal plasmids in somatic cells would need several rounds of cell culture. (4) mRNA-based non-integrative method: mRNA-carrying pluripotent genes are used for iPSC generation. mRNA reprogramming technology is the most unambiguously footprint free and genomic integration free for iPSC generation. However, conventional mRNA transcripts exhibit some disadvantages such as instability, immune activation, and difficult delivery, which limit their applications and reduce reprogramming efficiency. In recent years, researchers have gradually developed modified mRNA transcripts, which enhance their stability, reduce immunogenicity, and improve their delivery. Therefore, in this review, we summarized the reprogramming applications of modified mRNA (modRNA) in iPSC generation and therapeutic applications of these iPSCs and functionally differentiated cells.

## 2. Messenger RNA

### 2.1. Natural mRNA

Transcription is a process that makes an RNA copy from a DNA template through RNA polymerase. The RNA copy or transcript carries information and can be translated as a polypeptide [15]. In eukaryotes, an RNA transcript needs some processing including splicing and incorporation of 5′ cap and 3′ ploy-A tail on their ends [16]. The structure of a mature eukaryotic mRNA is shown in Figure 1. A mature mRNA contains 5′ cap, poly-A tail, 5′UTR (untranslated region), 3′ UTR, and ORF (open reading frame). The 5′ cap is a modified guanine nucleotide and is added to the 5′ end of mRNA. It contains a 7-methylguanosine residue (m7G), which is linked to the first nucleotide of mRNA through a 5′-5′-triphosphate. 5′cap exhibits protection from RNases and stabilizes the mRNA, in addition to the recognition by the ribosome for protein translation [17,18]. The poly-A tail with an optimal length of 100–200 nucleotides is added to the 3′ end of mRNA through polyadenylate polymerase, which the process calls polyadenylation [19]. An abnormal length makes the mRNA structure unstable. Polyadenylation is the covalent linkage of a polyadenylyl moiety to mRNA. A poly-U tail is also found in some mRNA [20]. The poly-A tail facilitates mRNA protection from exonucleases-caused degradation [21,22]. Poly-A tail also contributes to the export of mRNA from nucleus into cytosol for protein translation. 5′ UTR and 3′ UTR are located before the start codon of mRNA and after the stop codon of mRNA, respectively. UTRs can facilitate mRNA stability, translational efficiency, and mRNA localization depends on their sequence [23,24,25]. UTRs aid the stability of mRNA because of various affinities for RNA-degrading enzymes (ribonucleases). UTRs may influence translation efficiency due to the competition of other proteins with ribosomes for binding to mRNA [26]. 3′ UTR functions as the cytoplasmic localization of mRNA, which contains sequences that allow RNA transcript to go to the translated region [27,28]. The open reading frame is a coding region, which is decoded and translated into proteins by ribosomes. The stability of mRNA is also associated with internal base pairs of coding regions [29,30].

### 2.2. Modified mRNA

In recent years, substantial modifications of mRNA have been investigated for their extensive applications. The structure of a modified mRNA (modRNA) is shown in Figure 1, including the 5′ cap, poly-A tail, 5′ UTR, 3′ UTR, and ORF. In in vitro transcription (IVT), the natural 5′ cap (m7G) has the risk of uncapping or becoming inactive mRNA because of competitive incorporation between m7G and GTP. Pasquinelli et al. found that m7G was bound to the first nucleotide of the reverse orientated mRNA by 3′-5′ phosphodiester interaction, and one-third of the natural 5′ cap was incorporated in the reverse orientation [31]. Such reverse caps are unlikely to be recognized by translation initiation factor 4E (eIF4E). In 2001, Stepinski et al. designed a novel cap analog (P(1)-3′-O,7-dimethylguanosine-5′ P3-guanosine-5′ triphosphate), called anti-reverse cap analogs (ARCAs), which are not able to incorporate in the reverse orientation and skip degeneration by Dcp2 (mRNA-decaying enzymes) [32]. These significantly enhance translation efficiency and modRNA stability [32,33,34,35,36]. The natural poly-A tail can aid mRNA stability, thus, the optimal length of 120–150 nucleotides can be added to modRNA in in vitro transcription. There are two methods for poly-A tail addition: encoding the poly-A tail from the template vector or enzymatically adding adenine nucleotides using recombinant poly(A) polymerase [35,37]. The natural UTRs can enhance the stability and translation efficiency of modRNA depending on their sequences. Therefore, highly stabilizing UTRs derived from α/β-globin genes are used for desired modRNA [38].

Unmodified mRNAs such as ssRNA or dsRNA can induce interferon production through the activation of Toll-like receptor 7 (TLR7) and TLR8, thus limiting their extensive applications due to highly immunogenic properties [39,40,41,42]. Substantial modification in mRNA was reported [43], whereas only a small subset of modified nucleotides such as 5-methyluridine (5mU), 5-methylcytidine (5mC), pseudouridine (ψU), N6-methyladenosine (6mA), N1-methylpseudouridine (1mψU), and 5-methoxyuridine (5moU) were demonstrated to reduce innate immune responses and enhance translation efficiency and stability [34,35,38,44,45,46]. In 2010, 5mC and ψU were first used in the IVT modRNA-based reprogramming by inducing transgene expression of Oct4, SOX2, KLF4, c-Myc, and Lin-28 in fibroblasts for iPSC generation. The modRNA showed high transfection efficiency and high protein expression, whereas immune responses and cytotoxicity did not occur in the host [47]. In 2015, 1mψU was first incorporated into modRNA, which can reduce innate immune responses through inhibiting TLR3 activation [48]. In 2016, 5moU-modified mRNA was first investigated and exhibited high protein expression and extensive half-life [49]. In 2017, the 5mC was reported to increase mRNA-binding affinity and mRNA export through recognition by the mRNA export adaptor ALYREF [50]. Taken together, modRNA improves the disadvantages of original IVT mRNA and displays some advantages such as highly transient protein expression, very low immunogenicity, stability, improved delivery, and no genomic integration [51,52,53]. It is not yet suitable for the long-term expression of proteins [53].

Based on the understanding of modified mRNA, it can be designed to be synthesized in vitro by the robust T7 RNA polymerase-mediated transcription from a linearized DNA template, which incorporates UTRs such as 5′ UTR containing a Kozak sequence and α-Globin 3′ UTR. ARCA, modified nucleotides (5-Methylcytidine-5′-Triphosphate and Pseudouridine (ψ)-5′-Triphosphate) and poly-A tail could be incorporated in the mRNA to enhance the stability and reduce the immune response of host cells. DNase I could be added to digest the DNA template after modRNA synthesis. Phosphatase could be added to remove the 5′ triphosphates at the end of the modRNA to further reduce innate immune response in mammalian cells. After purification, the modRNA could be diluted in the buffer to the desired concentration for further application.

## 3. modRNA Delivery for Reprogramming

Some researchers used the modRNA strategy to overexpress pluripotent factors in somatic cells and subsequently reprogram cells into iPSCs. The modRNA-based reprogramming method contains the in vitro transfection of modRNA in somatic cells [47,52,54,55,56,57,58,59,60,61,62,63,64,65,66,67,68,69,70,71,72,73,74,75]. However, the cell membrane is a natural barrier for modRNA delivery. It is composed of a lipid bilayer including zwitterionic and negatively charged phospholipids [76]. There are two major delivery methods such as cationic lipid and electroporation used in modRNA-based reprogramming, listed in Table 1. The cationic lipids contain a positively charged head group and one or two hydrocarbon chains [77,78,79,80]. The positively charged head group controls the association between the lipid and phosphate backbone of the RNA and assists RNA condensation [81,82]. Cationic lipids are usually formulated with a helper lipid or a neutral co-lipid and result in liposomes with a positive surface charge when in water. The positively charged liposomes spontaneously interact with the negatively charged RNA and result in an RNA–liposome complex. This complex can fuse with cell membrane and enter the cytoplasm through endocytosis [83]. The modRNA of pluripotent factors immediately translate their target proteins and induce somatic cell reprogramming. The commercial RNAiMAX transfection reagent was most used in somatic cell reprogramming with modRNA, and other reagents such as Lipofectamine and Stemfect reagents were also used. Therefore, the cationic lipid-mediated delivery is a simple and fast method for transferring modRNA into somatic cells [47,54,56,57,58,59,60,61,62,63,64,65,66,68,69,70,71]. It was applied in somatic cell reprogramming such as adherent fibroblasts and mesenchymal stem cells [47,54,55,56,57,58,59,60,61,62,63,64,65,66,67,69,70,71].

Another modRNA delivery method used in reprogramming is electroporation [55,56]. This is a physical transfection method in which an electrical pulse is used to disturb the phospholipid bilayer of the cell membrane and results in transient pores in the membrane. The charged modRNA can be simultaneously driven across the cell membrane through the pores in a manner similar to electrophoresis when the electric potential moves across the membrane [84]. Electroporation is a highly efficient method for transferring exogenous RNA into many cell types and even blood cells that are difficult to transfect [85]. However, a high voltage pulse causes substantial cell death and only a small part of cells can survive through successful cell membrane repair. Thus, the electroporation method needs the use of greater quantities of cells and instruments compared with cationic lipid methods.

The switch of somatic cell reprogramming needs a substantial number of pluripotent factor expressions, which drive cell transformation into pluripotent states [2,7,86]. Therefore, modRNA-based reprogramming needs repeated transfections to induce overexpression of pluripotent proteins and further promote reprogramming due to the transient expression of modRNA transfections. In 2010, Warren et al. first used modRNAs including Oct4, SOX2, KLF4, c-Myc, and Lin-28 to reprogram various fibroblasts into iPSCs after 17 consecutive transfections of modRNA [47]. Although the modRNA-based reprogramming has a higher safety level with no genome insertion compared with integrative methods, the protocol is relatively complicated because of daily transfections over 2 weeks. Researchers gradually modified the protocol including a reduction in transfection numbers (1~5) and an increase in RNA dosages (43~5 μg) and successfully generated iPSCs using the modRNA transfection strategy [54,55,56,57,67,69]. The modRNA was also applied in cell differentiation; modified MYOD1 was able to induce fibroblasts to differentiate into footprint-free myoblasts. The transformation efficiency increases correlated with higher amounts of transfected modRNA [66].

In addition, another RNA delivery system such as graphene oxide-polyethylenimine (GO-PEI) was used for the reprogramming of somatic cells into iPSCs. Choi et al. used the delivery vehicle (GO-PEI) to avoid repetitive daily transfection of mRNA [67]. GO-PEI complexes were demonstrated to be very effective for transferring mRNA of pluripotent factors and protecting from mRNA degradation by RNase. GO-PEI/mRNA complexes-treated fibroblasts significantly enhanced the reprogramming efficiency and successfully generated human and rat iPSCs without daily transfection.

## 4. Modified mRNA-Based Reprogramming

The reprogramming efficiency can be as high as up to 4%, and colony formation begins around 14–18 days using modRNA-based technology. The modified mRNA was improved by highly transient protein expression, very low immunogenicity, stability, and no genomic integration. There are four combinations of various modifications that were used in the modRNA-based reprogramming, listed in Table 1. The modifications of mRNA (5mC, ψU, 5′ UTR containing a Kozak sequence, α-Globin 3′ UTR, Poly-A tail, ARCA) were frequently applied in nine studies [47,58,60,61,64,66,68,70,71]. Secondly, RNA modifications including 5mC, ψU, and ARCA were used in four studies [59,62,63,65].

The combinations of pluripotent factors such as OSKM and LIN28 were the most chosen for reprogramming modRNAs, and were used in seven studies. These modRNAs were usually mixed in a molar ratio of 3:1:1:1:1 (O:S:K:M:LIN28) [47,60,62,63,70,71]. NANOG was included in these five factors in four studies as well. Interestingly, microRNA also was combined with these modRNAs of pluripotent factors for enhancing the efficiency and kinetics of somatic cell reprogramming [65,70,71]. Lee et al. found that the numbers of daily modRNA transfections required for reprogramming were reduced from 17 [47] to 11 because of the combination with the microRNA cocktail (microRNA (miR)302a-d and miR367) [65]. The miRNAs-367/302s family of miRNAs has been demonstrated to induce pluripotency in fibroblasts [87]. The miR302/367 cluster can rapidly and efficiently reprogram mouse and human fibroblasts to a pluripotent state without exogenous transcription factors. The miR302/367-mediated somatic reprogramming occurs through the activation of Oct4 gene expression and suppression of Hdac2.

The modRNA-based reprogramming technology is utilized in various somatic cells such as fibroblasts [47,54,55,56,57,58,60,61,63,64,65,66,67,69,70,71], adipose-derived stem cells (ADSCs) [59], bone marrow-derived mesenchymal stem cells (BMSCs) [62], and amniotic fluid stem cells [68] for iPSC generation. Thirteen studies reported fibroblast reprogramming using modRNAs encoding pluripotent factors and generated disease iPSCs from patients with diseases such as cystic fibrosis [47], Huntington’s disease [56], DiGeorge syndrome [63], low-density lipoprotein receptor (LDLR) deficiency [64], and Down syndrome [70]. The β-thalassemia iPSCs were also generated from BMSCs using the modRNA platform [62].

The literature listed in Table 1 reported that modRNA-induced iPSCs differentiate into three germ layers in addition to five studies without evidence [54,55,57,60,71]. The modRNA-derived iPSCs (RiPSCs) also can further differentiate into other cell types in vitro such as myogenic cells [47], cardiomyocytes [58,63,68], hepatocytes [61,64], MSCs [64], and hematopoietic progenitors [62]. Taken together, RiPSCs have great promise in cardiac repair [58,63,68] and a new drug discovery [61,64], especially genetic correction of specific defects such as β-thalassemia [62,88,89] and low-density lipoprotein receptor (LDLR) deficiency familial hypercholesterolemia [64].

To date, modRNA-based reprogramming has been performed mostly on fibroblasts. However, blood cells have acted as a popular source for iPSC generation due to easy harvesting. The application of modRNA technology in the reprogramming of blood cells into iPSCs seems to be limited. Although the reprogramming efficiency of episomal plasmid-induced iPSC generation is lower than that of RNA systems, much literature has shown that various blood cells such as progenitors, lymphoblasts, monocytes, B, and T cells can be successfully reprogrammed using the episomal system combined with an electroporation delivery method [13]. In transfection numbers, the RNA system needs consecutive daily transfections for more than two weeks and lots of hands-on time, whereas the episomal system only needs a one-time plasmid transfection. Moreover, blood cells are well known to be difficult to transfect because of resistance with cationic reagents. Although the electroporation method can facilitate RNA delivery into somatic cells, repeat transfections can cause substantial death of somatic cells. Therefore, reduction in transfection numbers is required for modRNA-induced blood cell reprogramming to pluripotency. The increase in transfection efficiency can reduce transfection numbers by using an improved delivery method such as GO-PEI [67]. Additional reprogramming factors such as engineered chimeric pluripotent factors with extra transactivating domains can be incorporated into the modRNA system for enhancing reprogramming efficiency and then reducing transfection numbers. The co-transfection of microRNA also can synergize with pluripotent proteins to promote reprogramming and pluripotency [65,70,71]. In addition, the episomal system only prepares one plasmid including defined pluripotent factors such as OSKM, whereas the RNA system needs four separate OSKM plasmids to be prepared for in vitro transcription. Although the RNA system is more complicated and has a higher cost than the episomal system, RNA-based reprogramming holds great promise in the unambiguous footprint for future clinical-grade iPSC production. It is valuable for its use in overcoming the difficulty of reprogramming blood cell lineages with modRNA.

## 5. modRNA Applications in Cell Differentiation

iPSCs possess pluripotent properties and differentiate into any cell type in the body, which makes them potential therapeutic drugs in regenerative medicine. In addition to RiPSCs-differentiated therapeutic cells, modRNA can directly reprogram somatic cells into therapeutic cells bypassing iPSC generation. The modRNA-induced cell differentiation such as myoblasts, vascular endothelial cells, cardiomyocytes, endothelium, insulin-producing cells, neurons, and bone regeneration used in the therapeutic investigation are summarized in Table 2. Although modRNA-based reprogramming has advantages such as higher efficiency and safety than the viral methods, it usually requires repeat transfections of modRNA. In the application of pluripotent stem cell-derived therapeutic cells in vitro, the number of modRNA transfections required in the cell differentiation are reduced to 1~3 times. The MYOD modRNA can differentiate human RiPSCs into myogenic cells in vitro [47]. The vascular endothelial growth factor (VEGF) modRNA induces not only the vascular endothelial specification from ISL1^+^ heart progenitors in vitro but also promotes engraftment, proliferation, and survival of the human ISL1^+^ progenitors in vivo [90]. The modRNA encoding transcription factor ETV2 drove transient protein expression and was indeed sufficient to reprogram human pluripotent stem cells such as ESCs and iPSCs into hemogenic endothelial cells compared with lentivirus-mediated continuous ETV2 protein overexpression [91,92]. The ETV2 modRNA-induced hematoendothelial progenitors from human iPSCs were further cultured with GM-CSF, FGF-2, and UM171 to amplify myelomonocytic progenitors, followed by treatment with G-CSF and retinoic acid agonist Am580 to induce neutrophil differentiation [93,94]. The pancreatic-duodenal (PDX1) modRNA was also used in the driven differentiation of insulin-producing cells from human ESCs in vitro [95] and mouse pancreas-derived MSCs [96], which displays a promising approach for cell-based diabetic therapy. A cocktail of five transcription factors (NEUROG1, NEUROG2, NEUROG3, NEUROD1, and NEUROD2) as modRNAs can differentiate hPSCs into motor neurons in 7 days, analyzed by calcium imaging and electrophysiology [97]. In bone regeneration, the bone morphogenetic proteins (BMP-2) play a critical role in the development of bone and cartilage and induce osteoblast differentiation in a variety of cell types [98,99]. The modRNA encoding BMP-2 protein was transferred into BMSCs combined with biomaterials such as micro-macro biphasic calcium phosphate (MBCP) and resulted in cellular osteogenesis [100].

The modRNA-related cell differentiation has been applied in animal models including myocardial infarction and bone defect, which are summarized in Table 2. A myocardial infarction (MI), commonly known as a heart attack, occurs when blood flow decreases or stops to a part of the heart, resulting in injury to the heart muscle. This is usually caused by a blockage of one or more of the coronary arteries [101]. Intramyocardial injection of VEGF modRNA (100 μg/heart) resulted in the expansion and directed differentiation of endogenous heart progenitors and further improved heart function in a mouse myocardial infarction model [102]. In addition, the differentiation of heart WT1^+^ epicardial cells requires the activation of insulin-like growth factor 1 receptor (IGF1R). The intramyocardial injection of IGF modRNA drives epicardial adipose tissue formation after myocardial injury [103]. The intramyocardial delivery of Brachyury modRNA-induced cardiopoietic stem cells from adipose-derived stem cells (ADSCs) can improve cardiac performance and protected against decompensated heart failure [104]. A microencapsulated modified messenger RNA (M^3^RNA) platform was used for Brachyury modRNA delivery to achieve a nonintegrating and viral-free transfection [105]. In application for bone defect, the modRNA combined with biomaterial was coated on the titanium implants, which was designed to produce BMP-2 protein for bone regeneration. In the bone defect models such as rat femur and calvarial defects, BMP-2 modRNA combined with different biomaterials was implanted into defects and resulted in bone regeneration and the acceleration of bone healing [106,107,108,109].

In the application of transdifferentiation in vitro, the number of modRNA transfections required in the specification of cell fate ranged 1~14 times. There are two studies that used MYOD modRNA to generate myoblasts directly differentiated from fibroblasts or MSCs in vitro, suggesting a potentially clinically relevant source of autologous cells for cardiac repair [66,110]. The delivery technology C-Lipo including the polyarginine-fused heart-targeting peptide and lipofectamine complex was applied in the encapsulation of three modRNAs (Gata4, Mef2c, and Tbx5), which dramatically enhanced modRNA transfection and resulted in the direct reprogramming of cardiac fibroblasts toward cardiomyocytes [111]. Insulin-producing cells can be directly reprogrammed from pancreatic exocrine cells AR42J [112] by the transfection of PDX1 modRNA. In the application of neural transdifferentiation in vitro, a set of modRNA cocktails hold great promise in nerve regeneration and diseases. A cocktail of two transcription factors (SOX2 and PAX6) as modRNAs can transdifferentiate adult fibroblasts into neural precursor cells, which generate immature GABAergic or glutamatergic neuronal phenotypes in conjunction with astrocytes [113].

The modRNA-related transdifferentiation has been applied in animal models including ischemia and diabetes, which are summarized in Table 2. In applications for ischemia, a study showed that ETV2 modRNA combined with hypoxia (5% oxygen) for 14 days can produce functional endothelial progenitor cells (EPCs) that can form capillary-like structures from skin fibroblasts. The transfer of functional EPCs could improve hindlimb ischemia in a mouse model [114]. In application for diabetes, the transcription factor MafA (musculoaponeurotic fibrosarcoma oncogene homolog A) was applied for modRNA and was sufficient to drive β-cell transdifferentiation from human pancreatic duct-derived cells. The functional β cells can secret insulin and C-peptide in response to glucose. Transplantation of the β cells into diabetic SCID-beige mice mitigated hyperglycemia through their functional glucose-responsive insulin secretion [115].

## 6. Safety

To facilitate the translation of iPSC technology to clinical medicine, an unambiguous footprint reprogramming method is required. The mRNA-based reprogramming technology toward iPSC generation and transdifferentiation is the most unambiguously footprint-free, effective, and safe method. It eliminates bio-containment concerns associated with viral integrative vectors, which need to screen cells for several weeks to confirm that viral material has been completely removed during cell passaging. However, exogenous mRNA was demonstrated to induce innate immune responses through the activation of type I interferons when fibroblasts were transfected with mRNA in vitro [116]. The mRNA was also reported to elicit adaptive immune responses against cancer progress by intramuscular injection of carcinoembryonic antigen mRNA [117]. The exogenous mRNA causes inflammatory immune responses and suppresses mRNA replication due to triphosphorylated mRNA and double-stranded RNA, which can be recognized by TLR 3, 7, and 8. These TLRs initiate the expression of inflammatory cytokines such as type I interferons, IL-6, IL-12, and TNF-α [45]. The mRNA delivery into cells is through endocytosis and results in the RNA-containing endosomes in cytoplasm. These TLRs are expressed on endosomal membranes and recognize ssRNA [118].

In current years, some modifications of mRNA were found to reduce immune responses compared with unmodified mRNA. The activation of TLR 3, 7, 8, and retinoic acid-inducible gene I can be inhibited by 5mC, 6mA, 5mU, ψU, and 2TU [34,44,45,46]. RNA-dependent protein kinase-mediated the immune responses and translation suppression by phosphorylating α subunit of translation initiation factor 2 can be limited when mRNA contains modified nucleotides such as ψU and 5mC [119]. The activation of innate immunity-associated components such as interferon-induced enzymes 2′-5′-oligoadenylate synthetase and RNase L can be reduced by ψU [120]. When modified mRNAs were used combined with the B18R interferon inhibitor for somatic reprogramming, it was possible to keep daily modRNA transfections for two weeks without eliciting significant immune responses [47,121]. Although a very low level of inflammation may occur when modRNA and B18R were applied, this might facilitate reprogramming. Reports showed that deliberate control of TLR signaling-mediated inflammation is critical in the success of mRNA-induced somatic reprogramming [122,123,124,125].

On the other hand, modRNA delivery methods such as cationic lipid and electroporation were the most used for somatic reprogramming in vitro, shown in Table 1. However, electroporation utilizes high voltage pulses for modRNA delivery into cells and causes substantial cell death, and only a small number of cells can survive. Cationic lipids are also associated with cell toxicity, which disrupts the integrity of a membrane structure due to the detergent property [126]. In high concentrations, a lipoplex consisting of cationic lipid molecules can trigger cell lysis and necrosis [127]. Cationic groups may interact with cellular protein kinase C to induce cell toxicity [128]. Moreover, the modRNA-based reprogramming for iPSC generation needs repeat RNA transfections for more than two weeks, which may cause a lot of cell death. Therefore, an improved modRNA delivery method is required for somatic reprogramming. The GO-PEI was used to avoid repetitive daily transfection of modRNA and transfection was reduced to three times [67]. In addition, microRNA such as miR302/367 was also applied to facilitate efficient reprogramming and to reduce transfection numbers [65,70,71].

The modRNA technology has also been applied in myocardial infarction animal models, which can improve cardiac dysfunction and long-term survival when VEGF modRNA are delivered with a lipid carrier [102]. However, a lipid-based delivery system causes infusion-associated hypersensitivity reactions, tissue injury, and local innate immune responses [129,130,131]. Recently, reports demonstrated that modRNA can be expressed in the heart without using lipid carriers [132] and can even improve cardiac function after intracardiac injection of VEGF modRNA [133]. In addition, although modRNA technology is a rapid and effective method to produce target proteins in vitro and in vivo, the protein levels encoded by modRNA cannot be precisely predicted. This weak point may be improved through the repeated administration of different dosages of modRNA for predicting protein concentrations.

## 7. Conclusions

Exogenous proteins can drive somatic cell reprogramming and transdifferentiation for iPSC-related medicine. Therefore, a safe and effective method is required for use in research and medicine. Initially, the most common method for exogenous protein expression depended on viral integrative methods, which have the risk of genomic mutation through the integration of transgenes into the host genome. The modified mRNA of in vitro transcription provides non-integrative and footprint technology for target protein expression. modRNA can directly drive specific cell fate and cell reprogramming from various somatic cells. Moreover, a modRNA cocktail can be designed to simultaneously induce multiple different proteins in somatic cells. The in vitro transcription has benefits for greater flexibility, control over stoichiometric ratios, and dose titrations. The modRNA-based reprogramming and transdifferentiation hold great promise in iPSC-related regenerative medicine and disease modeling. Schematic overview of modRNA applications in cell fate conversion and potential therapies of modRNA-induced cells is shown in Figure 2.

## Figures and Tables

**Figure 1 ijms-22-08148-f001:**
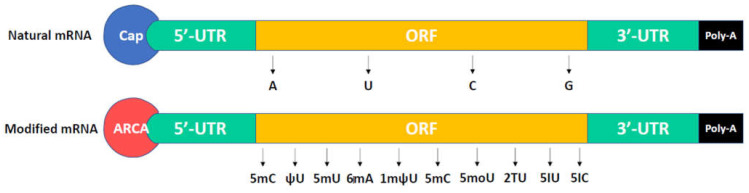
The structure of a mature eukaryotic mRNA and a modified mRNA. The anti-reverse cap analogs (ARCAs) and modified nucleotides such as 5-methyluridine (5mU), 5-methylcytidine (5mC), pseudouridine (ψU), N6-methyladenosine (6mA), N1-methylpseudouridine (1mψU), 5-methoxyuridine (5moU), 2-thiouridine (2TU), 5-iodouridine (5IU), and 5-iodocytidine (5IC) are usually used in modified mRNA.

**Figure 2 ijms-22-08148-f002:**
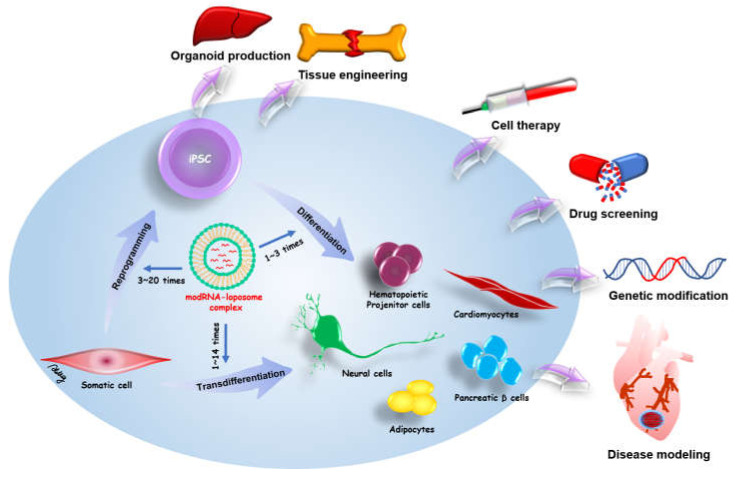
Schematic overview of modRNA applications in cell fate conversion and potential therapies of modRNA-induced cells. modRNA can be applied for somatic reprogramming to pluripotency (transfection numbers: 3~20 times) and stem cell differentiation to therapeutic cells (transfection numbers: 1~3 times). modRNA also can induce the transdifferentiation of somatic cells to distinct therapeutic cells (transfection numbers: 1~14 times). In addition, one direct injection of therapeutic modRNAs to defective organs may promote tissue regeneration and repair.

**Table 1 ijms-22-08148-t001:** Summary of reported iPSC generation using modRNA-based reprogramming.

Cell Sources	modRNA	Transfection Methods	Transfection Numbers	Total modRNA	Modifications	Differentiation to Three Germ Layers	Further Differentiation	References
BJ human neonatal foreskin fibroblasts, MRC-5 human fetal lung fibroblasts, Detroit 551 human fetal skin fibroblasts, dH1f fibroblasts, and skin cells of a cystic fibrosis patient	KLF4, c-MYC, OCT4, SOX2, LIN28	Cationic lipid	17	20 μg in 6-well plate; 136 μg in 10 cm dish	5mC, ψU, 5′ UTR containing Kozak sequence, α-Globin 3′ UTR, Poly-A tail, ARCA	Yes	Myogenic cells	[47]
Human foreskin fibroblasts	OCT4, LIN28, SOX2, NANOG	Cationic lipid	5	20 μg	Poly-A tail, ARCA, IRES sequence	No	N/A	[54]
Human fetal skin fibroblasts (HuF1), human embryonic lung fibroblasts (MRC5), human foreskin fibroblasts (HFF)	OCT4, SOX2, c-MYC, KLF4, SV40 large T (LT)	Electroporation	1	43 μg	Poly-A tail, ARCA, 5′ and 3′ UTRs of Xenopus b-globin	No	N/A	[55]
Human foreskin, adult Huntington fibroblasts, and adult skin fibroblasts of healthy donors	OCT4, NANOG, KLF4, c-MYC, SOX2, hTERT	Electroporation and Lipofectamine 2000	4	12 μg	Poly-A tail, Cap	Yes	N/A	[56]
Mouse embryonic fibroblasts (MEF)	OCT4, SOX2, KLF4, c-MYC	Cationic lipid	3	12 μg	Poly-A tail, ARCA	No	N/A	[57]
BJ neonatal fibroblasts, HDF-f fetal fibroblasts, HDF-n neonatal fibroblasts, HDF-a adult fibroblasts, and XFF xeno-free neonatal fibroblasts	OCT4, SOX2, KLF4, c-MYC-T58A, LIN28, NANOG	RNAiMAX	9	9 μg	5mC, ψU, 5′ UTR containing Kozak sequence, α-Globin 3′ UTR, Poly-A tail, ARCA	Yes	Cardiomyocytes	[58]
Adipose-derived mesenchymal stem cells of a 50-year-old patient	OCT4, KLF4, SOX2, LIN28, c-MYC	RNAiMAX	18	9.6 μg	5mC, ψU, Cap	Yes	N/A	[59]
Primary human fibroblasts	KLF4, c-MYC, OCT4, SOX2, LIN28, NDG	Cationic lipid	14	14 μg	5mC, ψU, 5′ UTR, 3′ UTR, Poly-A tail, ARCA	No	N/A	[60]
Human newborn foreskin fibroblasts	OCT4, KLF4, SOX2, LIN28, c-MYC	RNAiMAX	17	74.8 μg (Stemgent)	5mC, ψU, 5′ UTR containing Kozak sequence, α-Globin 3′ UTR, Poly-A tail, ARCA	Yes	Hepatocytes	[61]
Human bone marrow-derived mesenchymal stromal cells from a patient with β-thalassemia	OCT4, KLF4, SOX2, c-MYC, LIN28	RNAiMAX	18	21.6 μg	5mC, ψU, Poly-A tail, ARCA	Yes	Hematopoietic progenitors	[62]
Human adult dermal fibroblasts (HUF1 and HUF58), GM13325 fibroblasts from a 9-day-old patient with DiGeorge Syndrome, BJ human fibroblasts	OCT4, KLF4, SOX2, c-MYC, LIN28	RNAiMAX	12	14.4 μg	5mC, ψU, ARCA	Yes	Cardiomyocytes	[63]
Skin fibroblasts from a patient with low-density lipoprotein receptor (LDLR) deficiency, familial hypercholesterolemia (FH)	OCT4, SOX2, KLF4, c-MYC, LIN28	RNAiMAX	20	23.5 μg	5mC, ψU, 5′ UTR containing Kozak sequence, α-Globin 3′ UTR, Poly-A tail, ARCA	Yes	Hepatocytes, mesenchymal cells	[64]
Human adult dermal fibroblasts	OCT4, SOX2, KLF4, c-MYC, LIN28, miR302a-d, miR367	Stemfect RNA Transfection reagent	11	11 μg	5mC, ψU, Cap	Yes	N/A	[65]
Human BJ fibroblasts	OCT4, SOX2, KLF4, c-MYC, NANOG, LIN28,	Stemfect RNA Transfection reagent	9	2.2 μg	5mC, ψU, 5′ UTR containing Kozak sequence, 3′ UTR, Poly-A tail, ARCA	Yes	N/A	[66]
Human adipose-derived fibroblasts (ADFs), rat ADFs, mouse embryonic fibroblasts (MEF)	OCT4, SOX2, KLF4, c-MYC, or mRNA extracted from cells overexpressing OSKM	Graphene oxide-polyethylenimine (GO-PEI)	3	6 μg	5′ UTR, 3′ UTR, Poly-A tail, Cap	Yes	N/A	[67]
Human amniotic fluid stem Cells (AFSC)	OCT4, SOX2, KLF4, c-MYC, LIN28	RNAiMAX	18	79.2 μg	5mC, ψU, 5′ UTR containing Kozak sequence, α-Globin 3′ UTR, Poly-A tail, ARCA	Yes	Cardiomyocytes	[68]
Goat embryonic fibroblasts (GEF)	OCT4, SOX2, KLF4, c-MYC	Lipofectamine 2000	5	5 μg	Poly-A tail, ARCA	Yes	N/A	[69]
Human primary fibroblasts from two healthy donors and a patient with Down syndrome	OCT4, SOX2, KLF4, c-MYC, LIN28A, NANOG, mWasabi miR367/302s	RNAiMAX	7	4.4 μg	5mC, ψU, 5′ UTR containing Kozak sequence, α-Globin 3′ UTR, Poly-A tail, ARCA	Yes	N/A	[70]
Human primary fibroblasts	OCT4, SOX2, KLF4, c-MYC, LIN28A, NANOG, mWasabi miR367/302s	RNAiMAX	7	7 μg	5mC, ψU, 5′ UTR containing Kozak sequence, α-Globin 3′ UTR, Poly-A tail, ARCA	No	N/A	[71]

Abbreviations: 5-methylcytidine (5mC); pseudouridine (ψU).

**Table 2 ijms-22-08148-t002:** Summary of modRNA-induced transdifferentiation used in therapeutic investigations.

Cell Sources	modRNA	Modifications	Transfection Methods	Transfection Numbers	Total modRNA	Differentiated Cell Types	Animal Models	Therapeutic Effects	References
modRNA-induced hiPSCs	MYOD	5mC, ψU, 5′ UTR containing Kozak sequence, α-Globin 3′ UTR, Poly-A tail, ARCA	RNAiMAX	3	3.6 μg	Myogenic cells	N/A	N/A	[47]
Human foreskin fibroblasts	MYOD	5mC, ψU, 5′ UTR containing Kozak sequence, 3′ UTR, Poly-A tail, ARCA	Stemfect RNA transfection reagent	4	1.2 μg	Myoblasts	N/A	MYOD1 modRNA can directly transdifferentiate human fibroblasts into myoblasts without a transgene footprint	[66]
Mouse fibroblasts and hMSCs	MYOD	5mC, ψU, ARCA	Lipofectamine 2000	3	0.75 μg	Skeletal myoblasts	N/A	Defining optimized properties of modRNA-based protein expression in adult stem cells and fibroblasts	[110]
hESC-derived ISL1^+^ heart progenitors	VEGF-A	5mC, ψU, 5′ UTR containing Kozak sequence, α-Globin 3′ UTR, Poly-A tail, ARCA	RNAiMAX	In vitro-2 In vivo-1	In vitro-2 μg In vivo-5 μg	Human Isl1+ vascular endothelial cells	N/A	VEGF-A modRNA promotes not only the endothelial specification but also engraftment, proliferation, and survival (reduced apoptosis) of the human Isl1^+^ progenitors in vivo	[90]
Heart WT1^+^ epicardial progenitors	VEGF-A	5mC, ψU, 5′ UTR containing Kozak sequence, α-Globin 3′ UTR, Poly-A tail, ARCA	RNAiMAX	In vitro-1 In vivo-1	In vitro-3 μg In vivo-100 μg/heart	Endothelial cells and cardiovascular cells	Mouse myocardial infarction model	Modified mRNA directs the fate of heart progenitor cells and induces vascular regeneration after myocardial infarction	[102]
Endogenous heart epicardial progenitors	IGF1	5mC, ψU, ARCA	N/A	1	100 μg/heart	Epicardial adipose tissues	Mouse myocardial injury	An IGF1R modRNA-induced pathway drives epicardial adipose tissue formation after myocardial injury	[103]
Human ADSCs	Brachyury	5mC, Poly-A tail, ARCA	Microencapsulated-modified-mRNA (M^3^RNA) technique	1	1.75 μg	Cardiopoietic stem cells	Mouse myocardial infarction	Intramyocardial delivery of Brachyury modRNA-induced cardiopoietic stem cells can improve cardiac performance and protect against decompensated heart failure	[104]
Cardiac fibroblasts	Gata4, Mef2c, Tbx5	5mC, ψU, Poly-A tail, ARCA	C-Lipo (polyarginine-fused heart-targeting peptide and lipofectamine complex)	14	16.8 μg	Cardiomyocytes	N/A	C-Lipo can enhance modRNA transfection and results in the direct reprogramming of fibroblasts into cardiomyocytes	[111]
hESCs	ETV2, GATA2	5mC, ψU, Poly-A tail, ARCA	Electroporation	2	7 μg	CD43^+^ hematopoietic cells	N/A	Transient expression of ETV2 and GATA2 is indeed sufficient to commit the hPSCs to blood fate	[91]
Human skin fibroblasts	ETV2	Poly-A tail, Cap	Electroporation	1	3 μg	Endothelial progenitor cells	Hindlimb ischemia model	ETV2 modRNA combined with hypoxia can produce functional EPCs from fibroblasts and improve mouse ischemia	[114]
Human iPSCs	ETV2	ψU, Poly-A tail, ARCA	TransIT-mRNA	1	0.2 μg	Hemogenic endothelium	N/A	ETV2 modRNA-induced hematoendothelial progenitors can differentiate into functional neutrophils in the presence of G-CSF and Am580	[93,94]
Human iPSCs	ETV2	5′ UTR, 3′ UTR, Poly-A tail, Cap	Electroporation or RNAiMax	1	0.6 μg	Endothelial cells	N/A	Direct differentiation of human iPSCs into endothelial cells via transient modulation of ETV2 modRNA	[92]
hESCs	PDX1	5mC, ψU, Poly-A tail, ARCA	Electroporation	1	N/A	Insulin-producing cells	N/A	PDX1 modRNA can directly induce the transdifferentiation of insulin-producing cells	[95]
Mouse pancreas-derived MSCs	PDX1	5mC, ψU, Poly-A tail, ARCA	TransIT-mRNA	1	N/A	Insulin-producing cells	N/A	Mouse pMSCs can be transdifferentiated into functional glucose-responsive insulin-producing cells through transfecting PDX-1 modRNA	[96]
Pancreatic exocrine cells AR42J	PDX1, Ngn3, MafA	5mC, ψU, Poly-A tail, ARCA	Lipofectamine MessengerMAX	10	15 μg	Insulin-producing cells	N/A	Reprogramming of pancreatic exocrine cells into insulin-producing cells through modRNAs, represents a promising approach for cell-based diabetes therapy	[112]
Human pancreatic duct-derived cells	MafA	5mC, ψU, 5′ UTR containing Kozak sequence, α-Globin 3′ UTR, Poly-A tail, ARCA	jetPEI	7	8.4 μg	Insulin-producing cells	Diabetic SCID-beige mice	MafA modRNA can drive the reprogramming of human pancreatic duct-derived cells into functional insulin-secreting cells, and reverse diabetes	[115]
Human pluripotent stem cells	NEUROG1, NEUROG2, NEUROG3, NEUROD1, and NEUROD2	5mC, ψU, 5′ UTR, 3′ UTR, Poly-A tail, ARCA	Lipofectamine MessengerMAX	2	2 μg	Neurons	N/A	The modRNA cocktail can differentiate hPSCs into motor neurons	[97]
Human adult fibroblasts	SOX2, PAX6	5′ UTR, 3′ UTR, Poly-A tail, Cap	Lipofectamine RNAiMAX	4	8 μg	Neural precursor cells	N/A	Direct conversion of human fibroblasts into neural precursor cells using modRNA	[113]
HumanBMSCs	BMP-2	5mC, ψU, Poly-A tail, ARCA	Branched PEI	1	25 μg	Bone regeneration	Rat calvarial bone defect model	Scaffolds loaded with BMP-2 modRNA can enhance bone regeneration	[106]
Rat mesenchymal stem cells	BMP-2	5mC, 2TU, Poly-A tail, ARCA	C12-EPE	1	2.5 μg	Bone regeneration	Rat femur defect model	Delivering hBMP-2 modRNA to a femur defect can result in new bone tissue formation	[107]
Rat mesenchymal stem cells	BMP-2	5mC, 2TU, Poly-A tail, Cap	Proprietary lipid	1	2.5 μg	Bone regeneration	Rat femur defect model	BMP-2 modRNA-loaded collagen sponges can induce bone regeneration	[108]
HumanBMSCs	BMP-9, BMP-2	5mC, ψU, Poly-A tail, ARCA	PEI	1	50 μg	Bone regeneration	Rat calvarial bone defect model	BMP-9 modRNA can induce increased connectivity density of the regenerated bone compared with BMP-2 modRNA	[109]
Rat BMSCs	BMP-2	5mC, 2TU, Poly-A tail, ARCA	DF-gold	1	1 μg	Osteogenesis	N/A	The micro-macro biphasic calcium phosphate (MBCP) granules synergistically enhance the hBMP-2 modRNA-induced osteogenic pathway	[100]

Abbreviations: 5-methylcytidine (5mC); pseudouridine (ψU); 2-thiouridine (2TU).

## Data Availability

Not applicable.

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
