# Peer review of "Application of Modified mRNA in Somatic Reprogramming to Pluripotency and Directed Conversion of Cell Fate"

_ijms, 2021, doi:10.3390/ijms22158148_

Round 1

Reviewer 1 Report

In my opinion, additional references are required to support Authors conclusions. Moreover, Authors need to reproduce figures of critical studies and thoughfully describe some results.

More references are required for the statements in section 2, particularly for the last sentence.

Recent reviews on mRNA delivery should be cited.

References are required to support the affirmations of lines 145, 150, 161, 162, 164, 170, 176, 223, 233.

There seems to be an error in Tables 1 & 3: mg should be µg.

Line 206,the 9 papers should be cited, same for lines 207 and 230.

TISU are sequence-optimized mRNAs not modRNA; theses references do not fit in this review.

In conclusion, Authors need to expand the scope of the review to other applications of modRNA such as protein replacement therapy.

Author Response

Reviewer-1

In my opinion, additional references are required to support Authors conclusions. Moreover, Authors need to reproduce figures of critical studies and thoughfully describe some results.

Ans: Thank you very much for your comments. I have added additional references in the manuscript according to the comments below and added new contents in line 144~153, 248~273, and 358~403. I have also drawn a new Figure 2 in the manuscript (line 420~431).

Figure 2. Schematic overview of modRNA applications in cell fate conversion and potential therapies of modRNA-induced cells. modRNA can be applied for somatic reprogramming to pluripotency (transfection numbers: 3~20 times) or and stem cell differentiation to therapeutic cells (transfection numbers: 1~3 times). modRNA also can induce the transdifferentiation of somatic cells to distinct therapeutic cells (transfection numbers: 1~14 times). In addition, one direct injection of therapeutic modRNAs to defective organs may promote tissue regeneration and repair.

More references are required for the statements in section 2, particularly for the last sentence.

Ans: Thank you very much for your comments. I have added new references in section 2 (line 74, 75, 81, 83, 86, 90, 93, 94, 142, 143).

Recent reviews on mRNA delivery should be cited.

Ans: Thank you very much for your comments. I have added the current reviews about RNA delivery in line 157, 158.

References are required to support the affirmations of lines 145, 150, 161, 162, 164, 170, 176, 223, 233.

Ans: Thank you very much for your comments. I have added the relevant literature for these sentences (Line 145 to 158; line 150 to 162; line 161 to 173; line162 to 175; line 164 to 177; line 170 to 183; line176 to 188; line 223 to 233~235; line 233 to 243~244).

There seems to be an error in Tables 1 & 3: mg should be µg.

Ans: Thank you very much for your comments. Apology for this. This may be due to a problem with the uploaded Word file. I have changed all mg to ug in Table1 & 3.

Line 206,the 9 papers should be cited, same for lines 207 and 230.

Ans: Thank you very much for your comments. I have cited references for line 206 (new line 216: 9 papers), line 207 (new line 217: 4 papers), and line 230 (new line 241: 5 papers).

TISU are sequence-optimized mRNAs not modRNA; theses references do not fit in this review.

Ans: Thank you very much for your comments. I have deleted these references related to TISU in the manuscript.

In conclusion, Authors need to expand the scope of the review to other applications of modRNA such as protein replacement therapy.

Ans: Thank you very much for your comments. Apology for the confusing title. According to Reviewer 2’s comment, the manuscript title was changed to “Application of modified mRNA in somatic reprogramming to pluripotency and directed conversion of cell fate” in accordance with the manuscript content focusing on the somatic cell reprogramming and transdifferentiation. However, the reviewer points out a good opinion and I will review applications of modRNA in protein replacement therapy for another manuscript in the future.

Reviewer-2

This review specifically addresses the use of modified mRNA in cell reprogramming and transdifferentiation. As mRNA-based therapies in general have recently been in the spotlight, this is a timely review. Although interesting and well documented, there is some off-topic content, and some key points that should be discussed are missing.

Major

1 / The review goes behind cell reprogramming as it also deals with cell differentiation and even transdifferentiation. This is welcome. Thus, because the general issue here is to manipulate the fate of cells using exogenous mRNA, the title should be changed in accordance with this more general topic.

Ans: Thank you very much for your comments. I have revised the title to “Application of modified mRNA in somatic reprogramming to pluripotency and directed conversion of cell fate” according to the comment.

2 / For the reader's interest, add a few words on general methods used to produce modified mRNAs: IVT? Complete chemical synthesis? I understand that some companies are specialized in the production of mRNA? This could be added for example in chapter 2.

Ans: Thank you very much for your comments. I have added a new paragraph about modRNA synthesis in vitro in the end of chapter 2 (line 144~153).

3 / Table 2 and the part of the text discussing the differentiation capacities of iPS cells reprogrammed into mod RNA can be reduced to a few lines indicating that these iPS display the normal characteristics of pluripotency. It is not necessary to describe precisely what has been achieved with these cell lines, as these properties would be expected from genuine iPSC lines. Therefore, I strongly recommend removing Table 2 and lines 233-253.

Ans: Thank you very much for your comments. I have deleted these sentences to one sentence (line 244~247) and removed table 2.

4 / Source of the cells. It is surprising that no cellular reprogramming of blood progenitors by modRNA is reported, because reprogramming of blood progenitors has become the gold standard of cell reprogramming. Can the author comment on this particularity?

Ans: Thank you very much for your comments. I have added new comments in line 248~273 of manuscript.

5 / I would separate RNA-driven pluripotent stem cell differentiation from cell fate conversion (aka transdifferentiation). The first is a natural process that can be aided by modRNA, the second is an artificial process. Also, still in this chapter 5, I would separate the in vitro applications from the in vivo applications in separate paragraphs.

Ans: Thank you very much for your comments. I have separated modRNA-driven cell differentiation in vitro (line 275~305) from transdifferentiation in vitro (line 326~340) in different paragraphs. I have also separated the in vivo applications of cell differentiation (line 306~325) from the in vivo applications of transdifferentiation (line 341~351) in different paragraphs.

6 / Security. An important part of the safety chapter (Chapter 6) is devoted to describing the safety of in vivo administration of EPO modRNAs. I think this is off-topic and should be deleted (lines 332-347). On the other hand, certain aspects are missing in this chapter, such as the danger of activating the innate immune system, either in vitro (leading to cell death) or in vivo (leading to inflammation, see the history of the development of modRNA vaccines): this could be developed in this chapter and complement what has already been introduced in chapter 2.

Ans: Thank you very much for your comments. I have deleted these sentences (line 332~347) in the manuscript. New contents about risks of activating immune responses and cell death have been added in line 358~403 of manuscript.

Minor

- Beginning of chapter 4: lines 197-201: redundant with what has already been said before: can be deleted to lighten the text.

Ans: Thank you very much for your comments. I have deleted these sentences in line 197-201.

- English is generally correct, but some points need to be corrected

Ans: Thank you very much for your comments. I have revised the manuscript by the MDPI English editing.

Reviewer 2 Report

This review specifically addresses the use of modified mRNA in cell reprogramming and transdifferentiation. As mRNA-based therapies in general have recently been in the spotlight, this is a timely review. Although interesting and well documented, there is some off-topic content, and some key points that should be discussed are missing.

Major

1 / The review goes behind cell reprogramming as it also deals with cell differentiation and even transdifferentiation. This is welcome. Thus, because the general issue here is to manipulate the fate of cells using exogenous mRNA, the title should be changed in accordance with this more general topic.

2 / For the reader's interest, add a few words on general methods used to produce modified mRNAs: IVT? Complete chemical synthesis? I understand that some companies are specialized in the production of mRNA? This could be added for example in chapter 2.

3 / Table 2 and the part of the text discussing the differentiation capacities of iPS cells reprogrammed into mod RNA can be reduced to a few lines indicating that these iPS display the normal characteristics of pluripotency. It is not necessary to describe precisely what has been achieved with these cell lines, as these properties would be expected from genuine iPSC lines. Therefore, I strongly recommend removing Table 2 and lines 233-253.

4 / Source of the cells. It is surprising that no cellular reprogramming of blood progenitors by modRNA is reported, because reprogramming of blood progenitors has become the gold standard of cell reprogramming. Can the author comment on this particularity?

5 / I would separate RNA-driven pluripotent stem cell differentiation from cell fate conversion (aka transdifferentiation). The first is a natural process that can be aided by modRNA, the second is an artificial process. Also, still in this chapter 5, I would separate the in vitro applications from the in vivo applications in separate paragraphs.

6 / Security. An important part of the safety chapter (Chapter 6) is devoted to describing the safety of in vivo administration of EPO modRNAs. I think this is off-topic and should be deleted (lines 332-347). On the other hand, certain aspects are missing in this chapter, such as the danger of activating the innate immune system, either in vitro (leading to cell death) or in vivo (leading to inflammation, see the history of the development of modRNA vaccines): this could be developed in this chapter and complement what has already been introduced in chapter 2.

Minor

- Beginning of chapter 4: lines 197-201: redundant with what has already been said before: can be deleted to lighten the text.

- English is generally correct, but some points need to be corrected

Author Response

Reviewer-1

In my opinion, additional references are required to support Authors conclusions. Moreover, Authors need to reproduce figures of critical studies and thoughfully describe some results.

Ans: Thank you very much for your comments. I have added additional references in the manuscript according to the comments below and added new contents in line 144~153, 248~273, and 358~403. I have also drawn a new Figure 2 in the manuscript (line 420~431).

Figure 2. Schematic overview of modRNA applications in cell fate conversion and potential therapies of modRNA-induced cells. modRNA can be applied for somatic reprogramming to pluripotency (transfection numbers: 3~20 times) or and stem cell differentiation to therapeutic cells (transfection numbers: 1~3 times). modRNA also can induce the transdifferentiation of somatic cells to distinct therapeutic cells (transfection numbers: 1~14 times). In addition, one direct injection of therapeutic modRNAs to defective organs may promote tissue regeneration and repair.

More references are required for the statements in section 2, particularly for the last sentence.

Ans: Thank you very much for your comments. I have added new references in section 2 (line 74, 75, 81, 83, 86, 90, 93, 94, 142, 143).

Recent reviews on mRNA delivery should be cited.

Ans: Thank you very much for your comments. I have added the current reviews about RNA delivery in line 157, 158.

References are required to support the affirmations of lines 145, 150, 161, 162, 164, 170, 176, 223, 233.

Ans: Thank you very much for your comments. I have added the relevant literature for these sentences (Line 145 to 158; line 150 to 162; line 161 to 173; line162 to 175; line 164 to 177; line 170 to 183; line176 to 188; line 223 to 233~235; line 233 to 243~244).

There seems to be an error in Tables 1 & 3: mg should be µg.

Ans: Thank you very much for your comments. Apology for this. This may be due to a problem with the uploaded Word file. I have changed all mg to ug in Table1 &3.

Line 206,the 9 papers should be cited, same for lines 207 and 230.

Ans: Thank you very much for your comments. I have cited references for line 206 (new line 216: 9 papers), line 207 (new line 217: 4 papers), and line 230 (new line 241: 5 papers).

TISU are sequence-optimized mRNAs not modRNA; theses references do not fit in this review.

Ans: Thank you very much for your comments. I have deleted these references related to TISU in the manuscript.

In conclusion, Authors need to expand the scope of the review to other applications of modRNA such as protein replacement therapy.

Ans: Thank you very much for your comments. Apology for the confusing title. According to Reviewer 2’s comment, the manuscript title was changed to “Application of modified mRNA in somatic reprogramming to pluripotency and directed conversion of cell fate” in accordance with the manuscript content focusing on the somatic cell reprogramming and transdifferentiation. However, the reviewer points out a good opinion and I will review applications of modRNA in protein replacement therapy for another manuscript in the future.

Reviewer-2

This review specifically addresses the use of modified mRNA in cell reprogramming and transdifferentiation. As mRNA-based therapies in general have recently been in the spotlight, this is a timely review. Although interesting and well documented, there is some off-topic content, and some key points that should be discussed are missing.

Major

1 / The review goes behind cell reprogramming as it also deals with cell differentiation and even transdifferentiation. This is welcome. Thus, because the general issue here is to manipulate the fate of cells using exogenous mRNA, the title should be changed in accordance with this more general topic.

Ans: Thank you very much for your comments. I have revised the title to “Application of modified mRNA in somatic reprogramming to pluripotency and directed conversion of cell fate” according to the comment.

2 / For the reader's interest, add a few words on general methods used to produce modified mRNAs: IVT? Complete chemical synthesis? I understand that some companies are specialized in the production of mRNA? This could be added for example in chapter 2.

Ans: Thank you very much for your comments. I have added a new paragraph about modRNA synthesis in vitro in the end of chapter 2 (line 144~153).

3 / Table 2 and the part of the text discussing the differentiation capacities of iPS cells reprogrammed into mod RNA can be reduced to a few lines indicating that these iPS display the normal characteristics of pluripotency. It is not necessary to describe precisely what has been achieved with these cell lines, as these properties would be expected from genuine iPSC lines. Therefore, I strongly recommend removing Table 2 and lines 233-253.

Ans: Thank you very much for your comments. I have deleted these sentences to one sentence (line 244~247) and removed table 2.

4 / Source of the cells. It is surprising that no cellular reprogramming of blood progenitors by modRNA is reported, because reprogramming of blood progenitors has become the gold standard of cell reprogramming. Can the author comment on this particularity?

Ans: Thank you very much for your comments. I have added new comments in line 248~273 of manuscript.

5 / I would separate RNA-driven pluripotent stem cell differentiation from cell fate conversion (aka transdifferentiation). The first is a natural process that can be aided by modRNA, the second is an artificial process. Also, still in this chapter 5, I would separate the in vitro applications from the in vivo applications in separate paragraphs.

Ans: Thank you very much for your comments. I have separated modRNA-driven cell differentiation in vitro (line 275~305) from transdifferentiation in vitro (line 326~340) in different paragraphs. I have also separated the in vivo applications of cell differentiation (line 306~325) from the in vivo applications of transdifferentiation (line 341~351) in different paragraphs.

6 / Security. An important part of the safety chapter (Chapter 6) is devoted to describing the safety of in vivo administration of EPO modRNAs. I think this is off-topic and should be deleted (lines 332-347). On the other hand, certain aspects are missing in this chapter, such as the danger of activating the innate immune system, either in vitro (leading to cell death) or in vivo (leading to inflammation, see the history of the development of modRNA vaccines): this could be developed in this chapter and complement what has already been introduced in chapter 2.

Ans: Thank you very much for your comments. I have deleted these sentences (line 332~347) in the manuscript. New contents about risks of activating immune responses and cell death have been added in line 358~403 of manuscript.

Minor

- Beginning of chapter 4: lines 197-201: redundant with what has already been said before: can be deleted to lighten the text.

Ans: Thank you very much for your comments. I have deleted these sentences in line 197-201.

- English is generally correct, but some points need to be corrected

Ans: Thank you very much for your comments. I have revised the manuscript by the MDPI English editing.

Round 2

Reviewer 2 Report

The author responded adequately to all my remarks.

Some minor English language spell check might be required, for instance page 4 line 150 "Phosphatase could be added to remove the 5’ triphosphates at the end of the modRNA" change 'could' by can.